# Circulatory *Endothelin 1-*Regulating RNAs Panel: Promising Biomarkers for Non-Invasive NAFLD/NASH Diagnosis and Stratification: Clinical and Molecular Pilot Study

**DOI:** 10.3390/genes12111813

**Published:** 2021-11-18

**Authors:** Reda Albadawy, Sara H. A. Agwa, Eman Khairy, Maha Saad, Naglaa El Touchy, Mohamed Othman, Mohamed El Kassas, Marwa Matboli

**Affiliations:** 1Gastroentrology, Hepatology & Infectious Disease Department, Faculty of Medicine, Benha University, Benha 13736, Egypt; naglaaeltoukhy@yahoo.com; 2Clinical Pathology Department, Molecular Genomics Unit of Medical Ain Shams Research Institute, School of Medicine, Ain Shams University, Cairo 11566, Egypt; sarakariem@gmail.com; 3Medicinal Biochemistry and Molecular Biology Department, Ain Shams University School of Medicine, Cairo 11566, Egypt; dr_emankhairy@yahoo.com; 4Biochemistry Department, Faculty of Medicine, Modern University for Technology and Information, Cairo 12055, Egypt; maha.saad@medicine.mti.edu.eg; 5Gastroenterology and Hepatology Section, Baylor College of Medicine, Houston, TX 77030, USA; mohamed.othman@bcm.edu; 6Endemic Medicine and Hepato-Gastroenterology Department, Faculty of Medicine, Helwan University, Helwan 11792, Egypt; m_elkassas@hq.helwan.edu.eg

**Keywords:** steatohepatitis, NAFLD, NASH, TNF, RNA, RNAs panel

## Abstract

Nonalcoholic fatty liver disease (NAFLD) is one of the major seeds of liver cirrhosis and hepatocellular carcinoma. There is no convenient reliable non-invasive early diagnostic tool available for NAFLD/NASH diagnosis and stratification. Recently, the role of cytosolic sensor, stimulator of interferon genes (STING) signaling pathway in pathogenesis of nonalcoholic steatohepatitis (NASH) has been evidenced in research. We have selected *EDN1/TNF/MAPK3/EP300/hsa-miR-6888-5p/lncRNA RABGAP1L-DT-206* RNA panel from bioinformatics microarrays databases related to STING pathway and NAFLD/NASH pathogenesis. We have used reverse-transcriptase real-time polymerase chain reaction to assess the expression of the serum RNAs panel in NAFLD/NASH without suspicion of advanced fibrosis, NAFLD/with NASH patients with suspicion of advanced fibrosis and controls. Additionally, we have assessed the diagnostic performance of the Ribonucleic acid (RNA) panel. We have detected upregulation of the *EDN1* regulating RNAs panel expression in NAFLD/NASH cases compared to healthy controls. We concluded that this circulatory RNA panel could enable us to discriminate NAFLD/NASH cases from controls, and also NAFLD/NASH cases (F1, F2) from advanced fibrosis stages (F3, F4).

## 1. Introduction

Liver diseases cause two million deaths per year worldwide; thus, they represent a universal health problem [1]. Nonalcoholic fatty liver disease (NAFLD) is a progressive chronic liver disease characterized by excess fat accumulation in the liver. NAFLD can progress to nonalcoholic steatohepatitis (NASH) and, eventually, liver cirrhosis and hepatocellular carcinoma (HCC) worldwide [2]. There are new promising predictors of NAFLD as combination of serum biomarkers that could help in early NASH diagnosis, but, unfortunately, with several well-known limitations. Although magnetic resonance imaging-derived proton density fat fraction is considered the most accurate for fatty liver diagnosis. The main concern in clinical practice is early detection of NASH [3].

NAFLD progression is attributed to many pathways, e.g., oxidative stress, endoplasmic reticulum stress, and Toll-like receptor-dependent release of cytokines [4]. The liver acts as a primary immune cornerstone with various innate immune cells. Upon exposure to different stress signals, these innate immune cells become activated, inducing the innate immune response and stimulating liver inflammation [5]. The cytosolic DNA induces the cyclic GMP-AMP synthase (cGAS)-stimulator of interferon genes (STING) pathway representing a critical signaling pathway of the innate immune system [6]. Metabolic stress, such as a high-fat diet, obesity, and insulin resistance may stimulate cGAS and the STING-IRF3-mediated inflammation. Dysregulation of STING could inhibit free fatty acid induced inflammatory response, lipid accumulation, and hepatocellular damage [7]. Lipotoxic stimulation affects downstream targets of cGAS-STING kinase, it induces the nuclear factor kappa beta (NF-kB) signaling to produce proinflammatory cytokines that activate macrophage to produce *TGF-b1* and *TNF-a* which in turn stimulate hepatic stellate cells leading to liver fibrosis in NASH [8].

In NASH, chronic state of sterile inflammation is established due to the existing damage associated molecular patterns (DAMPs) DAMPs such as hepatocyte-mobility group-1 (*HMGB1*) and free fatty acids (FFAs) are endogenous molecules released from damaged cells that activate TLRs with subsequent inflammation, autophagy, and apoptosis [9,10]. In cellular stress conditions, the Hepatocyte mobility group (HMGB1) is moved from the nucleus into the cytoplasm, where it can affect intracellular processes such as autophagy. *HMGB* can act as pro-inflammatory mediators [11]. Moreover, extracellular HMGB1 activates G protein coupled receptor (GPCR) and thus mediating liver injury in NAFLD [12,13]. HMGB is linked to advanced glycation end products receptors (RAGE) that induce inflammation in NAFLD via several GPCRs [14].

There is an interesting crosstalk between hepatocyte and liver macrophages. DAMP including GPCR bound toll-like receptor 4 TLR to activate nuclear factor *(NF)-κB* and *TNFα* secretion in Kupffer Cells (KC) [15]. A recent study reported that the mitochondrial DNA acts as a stimulator of IFN genes (STING) in Kupffer Cells (KCs) to activate *TNFα* and *IL-6* synthesis under the conditions of lipid overload [16].

The role of non-coding ncRNA in NAFLD progression has been discussed by numerous research groups, e.g., miRNAs [17,18,19] and lncRNAs [20,21,22]. The integrated mRNA miRNAs lncRNA regulatory networks may provide new early diagnostic biomarkers and therapeutic strategies [23].

Based on these data, we constructed an “mRNAs–miRNAs–lncRNAs” regulatory RNA network linked to hepatocyte-liver macrophage cross talk in NAFLD pathogenesis based on in biomarker filtration from public microarray databases. Then, we assessed NAFLD/NASH patients’ status versus control participants and measured the differential expression of the selected NAFLD-specific RNA signature in sera samples.

## 2. Results

### 2.1. Retrieval of Differentially Expressed mRNAs (DEG) from GEO Data Set

By normalization and analysis of the microarray dataset, a number of DEGs were identified in GSE33814 (Figure 1, Appendix A). The GSE33814 dataset contained 9969 DEGs were identified based on the appropriate cut-off. We used the Enrichr database for functional enrichment analysis DEGs between NASH, steatosis, and normal groups (Figure 1, Appendix A). Differentially expressed genes were clustered upon their correlation coefficients. The co-expression matrix was represented by a heatmap graphed by heatmap R (Version 3.6.3) built function. Beige color represents down regulation while brick-red color represents up regulation. On the x-axis, samples were graphed against genes expression on the y-axis. The left cluster showed NASH samples (Figure 1A). Then, key genes *EDN1, EP300, MAPK3, and TNF* were selected for the targeted network and validated by other GEO datasets (Appendix A) and other public databases to be related to STING signaling, cytokine response, and NAFLD/NASH pathogenesis (Appendix A). These selected genes were imported into string database for PPI network construction (Appendix A). Additionally, Enricher Tool highlighted that DEG were linked to acute inflammatory response, TNF and MAP kinase signaling as the top 10 items of gene ontology and of KEGG pathways (Figure 1B,C). Additionally, we used the DAVID Functional enrichment tool (https://david.ncifcrf.gov/tools.jsp, accessed on 15 October 2021), which revealed that validated biological function *of EP300* and *MAPK3* in cytokine response and the molecular function of the four selected genes in regulation of RNA transcription and MAP kinase signaling (Appendix A, DAVID G supplementary table). Then, the targeted miRNA were selected from Target scan, namely: has-miR-6888-5p could interact with four differentially expressed mRNAs identified above lately, we used mirwalk2 to predict the interaction between lncRNAs and miRNAs *RABGAP1L-DT-206*, was screened and interacting with the retrieved miRNA (Appendix A). Finally, (*EDN1/TNF/MAPK3/EP300/hsa-miR-6888-5p/lncRNA RABGAP1L-DT-206* RNAs panel was constructed.

### 2.2. Analysis of Biochemical and Clinical Parameters in NAFLD/NASH

A remarkable difference was observed among the study groups versus control groups as regards BMI, total cholesterol, LDL, HDL-cholesterol, total triglycerides, total bilirubin, direct bilirubin, ALT, AST, alpha fetoprotein, serum albumin, GGT fasting blood glucose, glycated hemoglobin (HbA1C), HOMA-IR, and albumin-creatinine ratio (*p* = 0.00). Additionally, a significant difference was found among the study groups regarding diabetes mellitus history (*p* = 0.00). On the other hand, there was no difference of significance regarding sex among the different study groups (Table 1).

### 2.3. Dysregulated mRNA/miRNA/lncRNA Axis Expression in NAFLD/NASH

The current study evaluated the differential expression of the selected RNAs panel among the different study groups through measuring the fold change value (RQ). In comparison to the control group, significant up-regulation of *EDN1* mRNA, *EP300* mRNA *MAPK3* mRNA, and TNF mRNA expression levels in NAFLD and NASH groups was observed. Similarly, the expression of *hsa-miR-6888-5p* miRNA and *lncRNA RABGAP1L-DT-206* were found to be significantly up-regulated in group A and group B in comparison to control group (*p* = 0.00) (Figure 2A–C).

### 2.4. Diagnostic Performance of RNAs Panel in NASH

The diagnostic performance of the dysregulated RNAs panel was assessed by ROC curve analysis among the different study groups. The resulting AUC and cutoff values were able to differentiate NAFLD/NASH cases from controls, with AUC = 0.841 for *TNF* mRNA, AUC = 0.871 for *MAPK3* mRNA, AUC = 0.839 for *EP300* mRNA, AUC = 0.797 for *EDN1* mRNA, AUC = 0.916 for *miR-6888-5p* miRNA and AUC = 0.844 for *lncRNA RABGAP1L-DT-206*. The best cutoff values were 2.05, 2.65, 2.15, 1.85, 1.97, and 4.8 for TNF mRNA, MAPK3 mRNA, EP300 mRNA, EDN1 mRNA, miR-6888-5p miRNA, and lncRNA RABGAP1L-DT-206, respectively. The estimated sensitivities were 82%, 88%, 83%, 87%, 91%, and 81% respectively, with estimated specificities of 81%, 73%, 80%, 70%, 77%, and 83%, respectively. The aforementioned results represent the potential RNAs panel that could discriminate NAFLD cases from controls compared to the current biochemical non-invasive parameters such as AST, ALT, and GGT. (Table 2, Figure 3A–E). Moreover, the combined RNAs panel sensitivity was 91% and the specificity was 73%.

Furthermore, the diagnostic performance of the dysregulated mRNA/miRNA/lncRNA network was also assessed by ROC curve to compare group A versus group B. The best cutoff values were 3.4, 3.6, 4.1, 2.3, 4.05, and 2.3 for *lncRNA RABGAP1L-DT-206, miR-6888-5p miRNA, EDN1 mRNA, EP300 mRNA, MAPK3 mRNA, and TNF* mRNA, and, respectively, with AUC equal to 0.944, 0.628, 0.648, 0.707, 0.729, and 0.727 for the same targets, respectively. The estimated sensitivities were 100%, 50.7%, 62.5%, 81.3%, 68.8%, and 75.3%, respectively, with estimated specificities of 79%, 69.6%, 64.5%, 58.9%, 58.9%, and 58.9%, respectively. The aforementioned results confirm the bioinformatics results and support the suggestion that the selected RNA panel could help in diagnosis and differentiation of NAFLD/NASH without suspicion of advanced fibrosis from NAFLD/NASH with suspicion of advanced fibrosis (Table 2, Figure 3F).

The dysregulated RNA panel expression was not only effective in diagnosis of NAFLD/NASH and its differentiation from controls, but also in comparing different scores of NAFLD scoring and different scores of fibrosis scoring. Increased expression of the selected RNAs panel were observed, with either the higher the score of NAFLD score or the higher the score of fibrosis score (Figure 4A–D).

### 2.5. Correlation Analysis and Multivariate Regression Analysis of NASH Predictors

In order to validate the correlation between the selected RNAs panel, statistical correlation analysis was performed using Spearman’s coefficient. A significant positive correlation was found between *miR-6888-5p miRNA and TNF mRNA, MAPK3 mRNA, EP300 mRNA, EDN1 mRNA and lncRNA RABGAP1L-DT-206*. Additionally, significant positive correlation was observed *between lncRNA RABGAP1L-DT-206 and EDN1* mRNA. (Figure 5A–F) Moreover, a multivariate regression analysis was carried out. TNF mRNA (*p* = 0.025), *MAPK3* mRNA (*p* = 0.034), and *lncRNA RABGAP1L-DT-206* (*p* = 0.05) were observed to be independent predictors of NASH besides ALT (*p* = 0.011) (Table 3).

## 3. Discussion

Herein, based on the involvement of STING pathway in NAFLD progression and NASH development, we constructed an mRNA/miRNA/lncRNA regulatory RNA network linked to hepatocytes/macrophage/cytokine cross talk via in silico data analysis. Afterwards, we have assessed the serum expression of the selected RNA network in NAFLD, NASH cases, and controls to evaluate its efficacy in prediction and early diagnosis of NASH.

STING, which is a part of the innate immunity signaling pathway, shares in connecting upstream DNA sensors to downstream factors [24]. It was suggested that STING and interferon regulatory factor 3 (IRF3), have a fundamental role in early alcoholic disease pathogenesis [25]. Additionally, activation of the STING-IRF3 pathway was suggested to increase hepatocytes injury and dysfunction in NAFLD through stimulating apoptosis and inflammation, and dysregulating glucose and lipid metabolism [7]. A high-fat diet (HFD)-induced mtDNA release in a mouse model resulted in the activation of the STING pathway, leading to a chronic inflammatory response [26].

The EDN family consists of three peptides, including *EDN1, EDN2, and EDN3; of them, EDN1* is the most important mitogen and immunomodulator. EDN1 can exert mitogenic effects by binding to its receptor type A (*EDNRA*) [27]. Additionally, *EDN1* causes potent vasoconstriction [28], being implicated in energy metabolism, wound healing, liver fibrosis, and portal hypertension [29]. Farina et al. have showed that double-stranded ribonucleic acid (dsRNA) stimulated the *EDN1* protein and mRNA, and *EDN1* activation is mediated by TLR3 [30]. In NASH, liver sinusoidal endothelial cells (LSECs) turn dysfunctional and acquire vasoconstrictive phenotype with the release of increased levels of vasoconstrictors such as endothelin-1 (*EDN1)* [31]. In agreement with our results, Degertekin et al. reported an increase in the *EDN1* level in NASH patients compared to NAFLD [32]. EP300 (P300) and its related paralog CREBBP are transcriptional co-activators and major lysine acetyltransferases. [33]. *EP300* is a fundamental player in cell proliferation, differentiation, and apoptosis, and cellular epigenetic modification through target protein and transcription factors acetylation [34,35]. EP300 is also of the key genes in innate immunity [36]. Iqbal et al. have showed that the inflammasome interacts with STING leading to *TBK1* and *IRF3* phosphorylation, and nuclear IFN-β induction [37]. Oral supplementation with branched-chain amino acids (BCAA) in liver cirrhotic patients suppressed the expression of EP300 and decreased the incidence of HCC [38] that agree with the differential expression of *EP300* among the study groups.

Mitogen-activated protein kinase 3 (*MAPK3*), or extracellular signal-regulated kinase 1 (ERK1) is an important signal transducing component in the *ERK/MAPK* signaling pathway. It also has a vital role in the activation of the ERK/MAPK signaling pathway to transduce downstream signals [39]. The *p3-MAPK* signaling pathway has been revealed to modulate the production of IFN-β through STING to abolish innate immunity responses [40]. Liang et al. reported that STING activation increased the expression of *CCL22* through the MAPK/AP-1 signaling pathway [41]. Significant upregulation was found in the *ERK1/2* pathway in liver tumors from Mito-Ob-mice, indicating the role of obesity in NASH development [42]. Afrin et al. reported that Le Carbone (LC) reduced the level of *p-ERK1/2* in NASH mice, thus preventing progression of NASH [43]. The above-mentioned studies align with our presented data.

The tumor necrosis factor α (*TNFα*) gene is located in the major histocompatibility complex (MHC), specifically in the class III region: about 250 kb centromeric of the HLA-B locus and 850 kb telomeric of HLA-DR [44]. TNF is a potent cytokine with several pro-inflammatory effects [45]. The activation of the STING pathway leads to TANK-binding kinase 1 (TBK1) triggering; with phosphorylation induction of both NF-κB pathway and interferon regulatory factor 3 (IRF3), these changes concomitantly increase the expression of TNF and type I interferon (IFN) [24]. The roles of TNF and ER stress in NASH development have been established [46]. Accordingly, we assessed the TNF expression in relation to different mechanisms in NASH development. In agreement with our results, Todoric et al. reported an increase in liver TNF mRNA with a high fructose diet resulting in steatohepatitis [47]. Additionally, Nakagawa et al. also documented an increase in TNF expression that promoted lipogenesis, NASH, and HCC development. Additionally, they proposed that the use of anti-TNF drugs could arrest NASH and its progression into HCC [48].

miRNAs play vital roles in many biological processes and their dysregulation is linked to NAFLD pathogenesis [49]. Aberrant profiles of miRs, e.g., *miRNA-122* and *miRNA-34a*, could accelerate the development of metabolic syndrome and NAFLD [50,51,52]. In the current study, we have assessed the expression *hsa-miR-6888-5p* as a retrieved epigenetic activator of the EDN1/TNF/MAPK3/EP300/panel, in agreement with the recent evidence that miRNAs could interact with the promoter and enhance gene expression through man miRNA-induced RNA activation [53,54]. To the best of our knowledge, *hsa-miR-6888-5p* has not been related to liver disease before.

Several studies have highlighted the crucial regulatory roles of lncRNAs in NASH initiation and progression [20]. The interactions between these lncRNAs may clear the complexity and genetic regulation in NASH development, with the potentiality to become biomarkers aiding in early diagnosis and NASH severity assessment [55]. *LncRNA MALAT1* was found to be upregulated in fibrotic liver tissue after carbon tetrachloride (CCL4) treatment [56]. Additionally, MALAT1 could promote insulin resistance and hepatic steatosis through increasing the stability of nuclear SREBP-1c [57]. Additionally, PVT1 lncRNA was found to be upregulated in fibrotic liver tissue [58]. In the current study, we have assessed the expression of *lncRNA RABGAP1L-DT-206* as the master regulator of the *EDN1/TNF/MAPK3/EP300/hsa-miR-6888-5p* panel. To the best of our knowledge, *lncRNA RABGAP1L-DT-206* has not been attributed to NASH before. We detected increased expression of *lncRNA RABGAP1L-DT-206* in NAFLD and NASH cases, with optimal cutoff values that could differentiate NAFLD/NASH cases from controls; and also to discriminate NAFLD/NASH without suspicion of advanced fibrosis from NAFLD/NASH with suspicion of advanced fibrosis cases (Figure 6).

## 4. Materials and Methods

### 4.1. Biomarker Filtration of mRNA-miRNA-lncRNA Panel from Public Microarray Database

The candidate genes of the present study were acquired from the GEO database (www.ncbi.nlm.nih.gov/geo/, accessed on 15 October 2021) [59]. The search was restricted to homo sapiens and the experimental articles that contained whole-gene expression data that differentiate between the NASH and normal control groups were included. As a result, The GSE89632 dataset was used [60]. Detailed parameters of the dataset are presented in Appendix A. The GSE89632 dataset represents a cross-sectional study that used hepatic gene expression on Illumina Microarray and compared 20 patients with simple steatosis, 19 with nonalcoholic steatohepatitis (NASH), and 24 controls (HC). Subsequently, microarray data from the GSE89632 was submitted to the online database repository GEO2R (https://www.ncbi.nlm.nih.gov/geo/geo2r/, accessed on 15 October 2021) to identify differentially expressed genes (DEGs) among the groups (Appendix A). A *p*-value of <0.05 was considered to indicate a statistically significant difference. Finally, gene ontology (GO) enrichment and pathway analyses of the retrieved 9969 DEGs were performed using Enrichr (http://amp.pharm.mssm.edu/Enrichr, accessed on 15 October 2021) [61]. The result was summarized in (Appendix A and Figure 1).

Afterwards, the integrated RNA panel was filtered and verified in three steps from other GEO datasets and other microarray databases:

(i) *Endothelin 1* (EDN1), *E1A Binding Protein P300* (*EP300*), *Mitogen-Activated Protein Kinase 3*(*MAPK3*), and *Tumor Necrosis factor Alpha (TNFα*) were verified based upon their correlation to a STING-related cytokine response and strong implication in NASH pathogenesis. The chosen messenger RNAs were also verified for their gene ontology and expression by using several public microarray databases; QuickGO (https://www.ebi.ac.uk/QuickGO/, accessed on 23 October 2021), and National Center of Biotechnology Information Gene (https://www.ncbi.nlm.nih.gov/gene, accessed on 23 October 2021) (Appendix A) and by literature reviews [16,17,18,19,20,21,22] to be related to cytokine and Cytosolic DNA-sensing pathway STING signaling pathway by KEGG (https://www.genome.jp/kegg/, accessed on 23 October 2021) (Appendix A). The four chosen genes were uploaded into the Search Tool for the Retrieval of Interacting Genes (STRING; version 11.0; http://stringdb.org, accessed on 23 October 2021) database to assess protein–protein cross talk (Appendix A) and DAVID functional enrichment tool to highlight their gene ontology in NAFLD/NASH progression (Appendix A).

(ii) We used Targetscan database to select miRNA that interact with the four selected mRNAs. It revealed that miR-6888-5p could target the selected mRNAs (Appendix A.). Additionally, miRPath database version 2 (https://mpd.bioinf.uni-sb.de/mirna.html?mirna=hsa-miR-6888-5p&organism=hsa, accessed on 23 October 2021) was used to carry out pathway enrichment analysis of *miR-6888-5p* that was linked to regulation of gene expression, RNA polymerase, and cell morphogenesis.

(iii) We used miRWalk 2.0; miRNA:ncRNA target tool (http://zmf.umm.uni-heidelberg.de/apps/zmf/mirwalk2/mir-mir-self.html, accessed on 23 October 2021) to predict the interaction between miRNA and *lncRNA. RABGAP1L-DT 206*(ENSG00000227373, ENST00000454467.1) was identified to be interacting with the chosen miR-6888-5p, and that was validated through Clustal Omega tool of The European Bioinformatics Institute (EMBL-EBI) (https://www.ebi.ac.uk/Tools/msa/clustalo/, accessed on 23 October 2021) (Appendix A).

All in all, (*EDN1, EP300, MAPK3 & TNFα)—(miR-6888-5p)—(RABGAP1L-DT-206)* RNA panel was constructed.

### 4.2. Study Subjects

A total of 200 participants were included in the current study: 60 cases NAFLD/NASH without suspicion of advanced fibrosis, 40 cases NAFLD/NASH with suspicion of advanced fibrosis, and 100 controls. The study cases were coming for medical assessment in Benha University Hospitals’ hepatology clinics from June 2020 to December 2020. Controls were receiving a routine health check in the hospital clinics. The Benha University ethical committee, faculty of medicine has approved the current study (approval number: MoHP0018122017, 1017), and all of the study population signed written informed consent before their participation.

NAFLD/NASH were diagnosed according to the following criteria [62]: no alcohol intake in the year preceding the study, clinical picture with confirmed steatosis by imaging modalities, exclusion of other liver diseases, e.g., schistosomiasis, viral hepatitis viral markers, and bilharzial antibodies detection, were performed and cases were excluded when positive to any of them.

Concomitantly, following fasting abdominal ultrasound (Acuson S2000, Siemens (Medical Solutions, Mountain View, CA, USA)) performed by 3 medical radiologists, the steatosis score was assessed, categorizing patients into 19 non-steatosis cases, 16 mild cases, 24 moderate cases, and 41 severe cases. Transient elastography (Fibroscan1) was used to assess the fibrosis score, categorizing patients into 34 mild liver scarring cases, 27 moderate liver scarring cases, 29 severe liver scarring cases, and 10 advanced liver scarring cases. Additionally, the NAFLD score was assessed in the different groups. Furthermore, controls were age and sex matched to the study cases, with negative viral markers and bilhariziasis, no alcoholic history intake, and normal liver function test, with confirmed normal imaging findings.

Blood samples were collected, further processed by 20 min centrifugation at 4000 rpm. The upper serum was collected and kept at −80 °C in a freezer for further usage.

Multifunctional biochemistry analyzer (AU680, Beckman Coulter Inc., Kraemer Blvd., Brea, CA 92821, USA was applied to assess the liver function tests, lipid profile, fasting blood glucose, HbA1C, and AFP (Appendix A). The fasting insulin levels, assessed by ELISA HOMA-IR, were calculated according to the formula: Fasting insulin (μU/L) × fasting glucose (nmol/L)/22.5.

### 4.3. Total RNA Extraction and Quantitative Real Time PCR (RT-qPCR)

RNA extraction from the sera samples was processed with miRNEasy extraction kit (Qiagen, Hilden, Germany) according to the manufacturer’s instructions. The quality of the purified RNA was measured by the Qubit TM ds DNA HS Assay Kit and Qubit TM RNA HS Assay Kit (Catalogue no. Q32851and Q32852, respectively, Invitrogen by Thermo Fisher Scientific, Eugene, OR, USA) on Qubit 3.0 Fluorimeter (Invitrogen by life technologies, Malaysia).

A total of 0.5 µg of RNA extracted from sera samples was used for reverse transcription using miScript II RT kit (Qiagen, Hilden, Germany; Cat no. 218161). Relative expression of the different targets was assessed using QuantiTect SYBR Green PCR Kit (Cat no. 204143, Qiagen, Hilden, Germany) for EDN1, EP300, MAPK3 and TNF genes, RT^2^ SYBR Green ROX qPCR Master mix (Cat no: 330500; Qiagen, Hilden, Germany) for lncRNA RABGAP1L-DT-206 and miScript SYBR Green PCR Kit (Cat no. 218073, Qiagen, Hilden, Germany) for hsa-miR-6888-5p miRNA on 7500 Fast System (applied Biosystems, Foster City, CA, USA) thermal cycler according to the manufacturer’s protocol. The list of used primers for quantitative RT-PCR is listed in Appendix A Gene expression levels were normalized to GAPDH, SNORD72. All samples were run in two replicates per experiment. Fold changes (Relative expression, RQ) were calculated according to 2^−ΔΔCt^ formula.

### 4.4. Statistical Analysis

The results were presented as mean ± SD for symmetrically distributed raw numerical data and median for non-parametric data using the software package of statistical analysis version 25 (SPSS version 25). Statistical analysis was performed using one-way ANOVA, chi-square test, and Spearman correlation. Regarding the predictive value of the selected panel in NASH diagnosis, the receiver operating characteristic (ROC) curve was used. Significance was set at *p* = 0.00 to *p* < 0.001.

## 5. Conclusions

Enlightened by the increasing data about the implication of STING signaling pathway in many diseases and the increasing prevalence of NASH without available reliable non-invasive diagnostic tool, we have retrieved a novel RNA panel from public microarray databases. The selected RNA panel is related to hepatocyte/liver macrophage/ STING pathway that could be potential noninvasive tool for diagnosis and early prediction of NASH in clinical pilot study. We reported upregulation of the *EDN1* regulating RNAs panel expression in NAFLD and NASH cases. Based on the diagnostic performance analysis of this RNAs panel, we concluded that the circulatory *EDN1* Regulating RNAs panel could enable us to discriminate NAFLD/NASH cases from controls, and also NAFLD/NASH cases with early (F1, F2) from advanced Fibrosis (F3, F4) (Figure 6).

## Figures and Tables

**Figure 1 genes-12-01813-f001:**
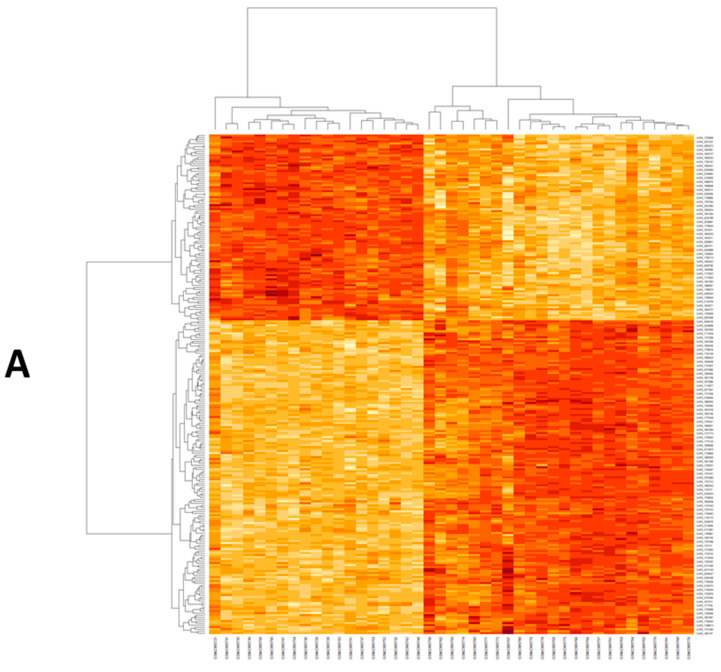
(**A**) Heat map of differentially expressed genes in GSE89632. The co-expression matrix was represented by a heatmap graphed by heatmap R (Version 3.6.3) built function. Beige color represents down regulation while brick-red color represents upregulation. (**B**) Top 10 items of Gene Ontology (Biological processes) for the retrieved DEGs according to *p* value obtained from Enrichr. (**C**) Top 10 items of KEGG pathways for the retrieved DEGs according to adjust *p* value obtained from Enrichr.

**Figure 2 genes-12-01813-f002:**
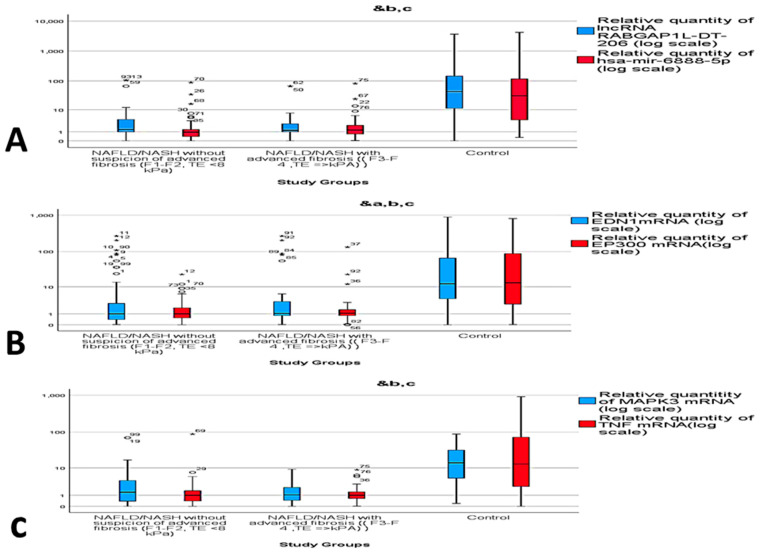
Relative expression of circulatory RNAs panel among the study groups. (**A**) lncRNA RABGAP1L-DT-206, hsa-miR-6888-5p, (**B**) EP300 mRNA, EDN1 mRNA and (**C**) TNF mRNA and MAPK3 mRNA. ^&^ Statistically significant difference by post Hoc (Turkey) test. ^a^ control vs. Group A, ^b^ control vs. Group B, ^c^ Group A vs. Group B. *, circle: represent outliers.

**Figure 3 genes-12-01813-f003:**
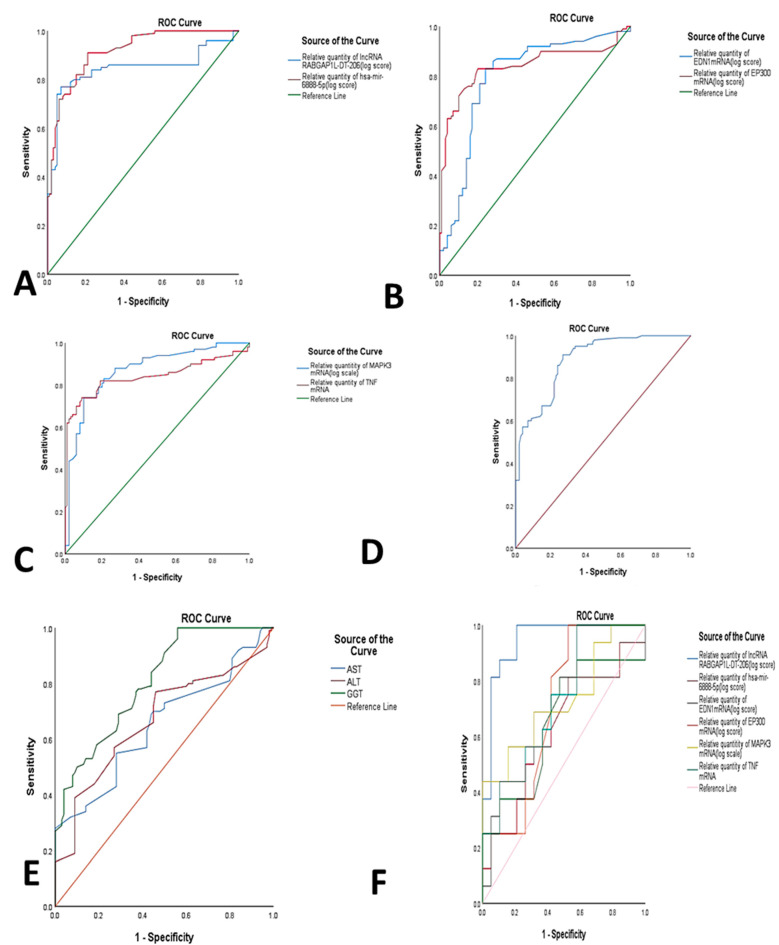
ROC curve analysis of (**A**) *lncRNA RABGAP1L-DT-206, hsa-miR-6888-5p* between NAFLD/NASH and controls; (**B**) *EP300 mRNA, EDN1* mRNA between NAFLD/NASH and controls; (**C**) *TNF mRNA, MAPK3* mRNA between NAFLD/NASH and controls. (**D**) Combined RNAs panel between NAFLD/NASH and controls (**E**) ALT, AST, GGT between NAFLD/NASH and controls; (**F**) *lncRNA RABGAP1L-DT-206, hsa-miR-6888-5p, EP300 mRNA, EDN1 mRNA, TNF mRNA;* and *MAPK3 mRNA* panel between group A and B.

**Figure 4 genes-12-01813-f004:**
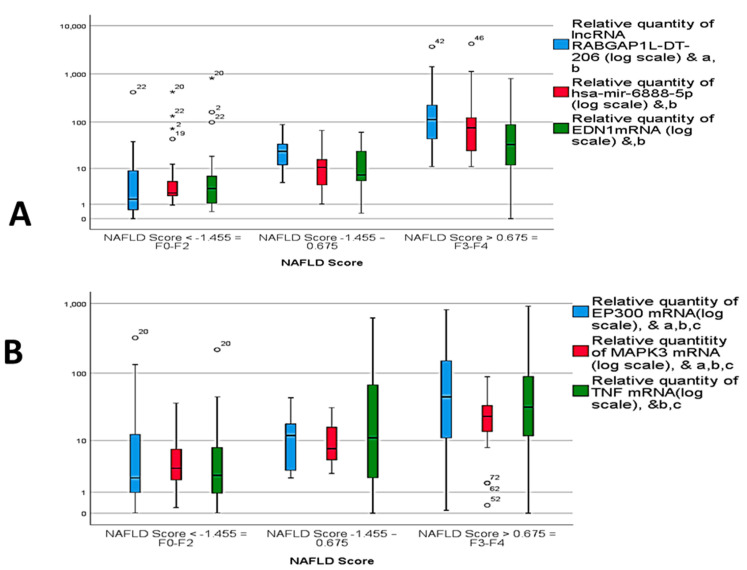
Relative expression of circulatory RNA panel among different NAFLD scores of (**A**) *lncRNA RABGAP1L-DT-206, hsa-miR-6888-5p and EDN1 mRNA;* (**B**) *TNF mRNA, MAPK3 mRNA, and EP300 mRNA* and statistically significant difference (*p* < 0.05) by Tukey post hoc test, a, b NAFLD score 1 vs. 3, c NAFLD score 2 vs. 3. (**C**) Relative expression among different fibrosis scores of *TNF mRNA, MAPK3 mRNA, and EP300 mRNA*, and (D) relative expression among different fibrosis scores of *lncRNA RABGAP1L-DT-206, hsa-miR-6888-5p, and EDN1* mRNA and statistically significant difference by post Hoc (Turkey) test. ^&^ statistically significant difference (*p* < 0.05) by Tukey post-hoc test, ^b^ Fibrosis stage 2 vs. Fibrosis stage 3, ^c^ Fibrosis stage 1 vs. Fibrosis stage 3, ^d^ Fibrosis stage 1 vs. Fibrosis stage 4, ^e^ Fibrosis stage 2 vs. Fibrosis stage 4, ^f^ Fibrosis stage 3 vs. Fibrosis stage 4.

**Figure 5 genes-12-01813-f005:**
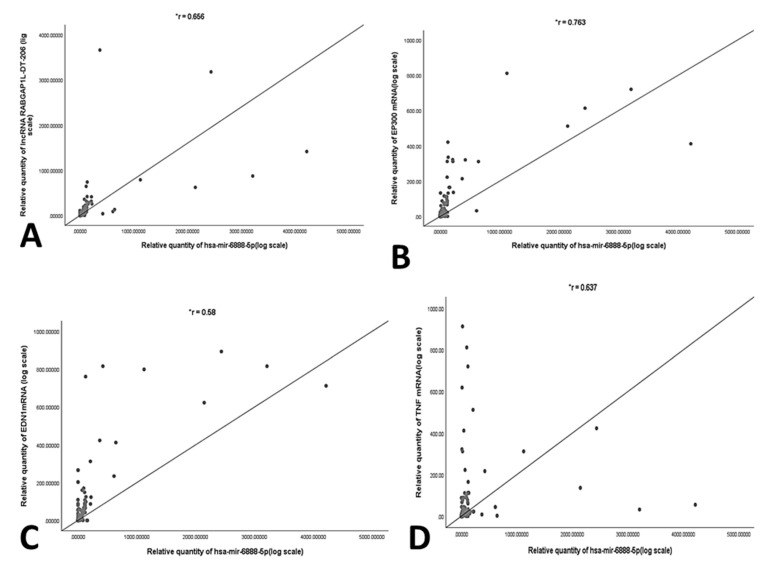
Correlation analysis using spearmen’s coefficient between (**A**) RQ of *RABGAP1L-DT-206 and hsa-miR-6888-5p*, (**B**) RQ of *hsa-miR-6888-5p* and *EP300* mRNA, (**C**) RQ of *hsa-miR-6888-5p* and *EDN1* mRNA, (**D**) RQ of *hsa-miR-6888-5p* and *MAPK3* mRNA, (**E**) RQ *of hsa-miR-6888-5p* and *TNF* mRNA, and (**F**) RQ of *RABGAP1L-DT-206 and EDN1* mRNA. *: significant *p* < 0.05.

**Figure 6 genes-12-01813-f006:**
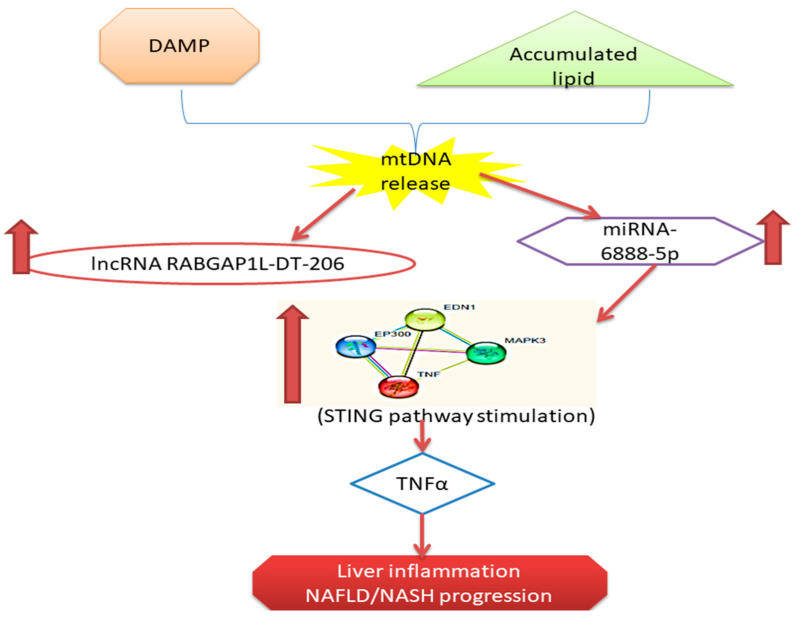
Summary and schematic presentation of the study findings.

**Table 1 genes-12-01813-t001:** Clinical and laboratory characteristics among the groups of the study.

Variable	Group A NAFLD/NASH without Suspicion of Advanced Fibrosis (F1–F2, TE < 8 kPa), *n* = 60	Group B NAFLD/NASH with Suspicion of Advanced Fibrosis (F3–F4, TE ≥ 8 kPa), *n* = 40	Group 3 Control *N* = 100	*p* Value
Sex				
male	39 (65%)	30 (75%)	64 (64%)	0.441
female	21 (35%)	10 (25%)	36 (36%)	
History of diabetes mellitus				0.00 **
positive	49 (81.7%)	34 (85%)	42 (42%)	
negative	11 (18.3%)	6 (15%)	58 (58%)	
Body mass index (kg/m^2^) BMI	35.5 ± 5.1	33.7 ± 6.7	25.9 ± 3.3	^a^ 0.00 ** ^b^ 0.00 ** ^c^ 0.134
Total cholesterol (mg/dL)	298.18 ± 59.6	289.4 ± 60.1	189 ± 85.9	^a^ 0.00 ** ^b^ 0.00 ** ^c^ 0.545
LDLc (mg/dL)	209.9 ± 49.5	199 ± 60.4	136.18 ± 66.3	^a^ 0.001 ** ^b^ 0.00 ** ^c^ 0.386
HDLc (mg/dL)	30.8.5 ± 9.09	27.6 ± 6.5	50.43 ± 20.8	^a^ 0.00 ** ^b^ 0.00 ** ^c^ 0.328
Total triglycerides (mg/dL)	270.3 ± 77.6	298.15 ± 58.4	179.7 ± 90.7	^a^ 0.00 ** ^b^ 0.00 ** ^c^ 0.106
albumin creatinine ratio	25.07 ± 4.2	23.5 ± 5.01	20.2 ± 6.9	^a^ 0.00 ** ^b^ 0.00 ** ^c^ 0.207
AST (IU/L)	71.2 ± 36.9	70.6 ± 41.2	51 ± 19.7	^a^ 0.00 ** ^b^ 0.002 ** ^c^ 0.993
ALT (IU/L)	46.3 ± 25.2	59.7 ± 44.8	34.3 ± 16.4	^a^ 0.00 ** ^b^ 0.007 ** ^c^ 0.012 *
Total bilirubin(mg/dL)	2.6 ± 0.9	3 ± 0.8	1.5 ± 1.2	^a^ 0.00 ** ^b^ 0.00 ** ^c^ 0.057
Direct bilirubin (mg/dL)	1.5 ± 0.66	1.7 ± 0.69	0.88 ± 0.39	^a^ 0.00 ** ^b^ 0.00 ** ^c^ 0.022 *
Albumin(g/dL)	2.5 ± 0.5	2.4 ± 039	3.23 ± 0.3	^a^ 0.00 ** ^b^ 0.00 ** ^c^ 0.065
Gamma glutammyl transferase (IU/L)	57.8 ± 39.9	65.6 ± 31.3	22.3 ± 21.7	^a^ 0.00 ** ^b^ 0.00 ** ^c^ 0.243
Alpha fetoprotein	180.5.9 ± 439	359 ± 433	18.0 ± 31.27	^a^ 0.004 ** ^b^ 0.00 ** ^c^ 0.012 *
Fasting blood glucose(mg/dL)	207.5 ± 83.3	179.3 ± 83.5	151.0 ± 87	^a^ 0.000 * ^b^ 0.106 ^c^ 0.194
Glycated hemoglobin HbA1c (%)	7.07 ± 1.09	7.8 ± 2.01	6.5 ± 2.7	^a^ 0.27 ^b^ 0.008 ** ^c^ 0.000 *
HOMA IR	12.66 ± 7.9	19.3 ± 6.8	5.0 ± 6.1	^a^ 0.000 ** ^b^ 0.00 ** ^c^ 0.000 *
NAFLD Score			---	--
NAFLD Score < −1.455 = F0–F2	27 (45%)	0 (0%)		
NAFLD Score −1.455 − 0.675	33 (65%)	2 (5%)		
NAFLD Score > 0.675 = F3–F4	0 (0%)	38 (95%)		
Fibrosis score			---	---
F0 to F1 Mild liver scaring	34 (56.7%)	0 (0%)		
F2: Moderate liver scarring	26 (46.3%)	0 (0%)		
F3: Severe liver scarring	0 (0%)	29 (72.5%)		
F4: Advanced liver scarring (cirrhosis)	0 (0%)	11 (27.5%)		
steatosis grading				
S1 mild steatosis	15 (25%)	0 (0%)		
S2 moderate steatosis	21 (35%)	4 (10%)		
S3 severe steatosis	5 (8.3%)	36 (90%)		
S4 non steatosis	19 (31.7%)	0 (0%)		

One way ANOVA test with post Hoc Turkey test was performed to assess the differences among the study groups. Abbreviation: AST = aspartate transaminase, ALT = alanine transaminase, BMI = body mass index, FBS = fasting blood sugar, GGT = Gamma glutamyl transferase, HDL-C = high density lipoprotein cholesterol, LDL-C = low density lipoprotein cholesterol, TE =Transient elastography, Kpa = kilopascal ^a^ control vs. Group A, ^b^ control vs. Group B, ^c^ Group A vs. Group B ** *p* < 0.01; * *p* < 0.05.

**Table 2 genes-12-01813-t002:** Diagnostic performance of the molecular parameters among the study groups.

Test Result Variable(s)	Area	Std. Error	Asymptotic Sig.	Asymptotic 95% Confidence Interval		Cutoff	Sensitivity	Specificity
				Lower Bound	Upper Bound			
**NAFLD/NASH vs. Control**								
*lncRNA RABGAP1L-DT-206*	0.844	0.031	0.000	0.82	0.905	4.8	81%	83%
*has-miR-mir-6888-5p*	0.916	0.019	0.000	0.879	0.953	1.97	91%	77%
*EDN1mRNA*	0.797	0.033	0.000	0.731	0.862	1.85	87%	70%
*EP300 mRNA*	0.839	0.031	0.000	0.779	0.900	2.15	83%	80%
*MAPK3 mRNA*	0.871	0.026	0.000	0.820	0.921	2.65	88%	73%
*TNF mRNA*	0.841	0.031	0.000	0.781	0.901	2.05	82%	81%
Combined RNAs	0.888	0.022	0.000	0.923	0.845	3.25	91%	73%
AST	0.653	0.039	0.000	57	0.577	0.729	55%	72%
ALT	0.669	0.039	0.000	27	0.593	0.745	57%	73%
GGT	0.806	0.030	0.000	39.5	0.748	0.864	66%	71%
**Group A vs. Group B**								
*lncRNA RABGAP1L-DT-206*	0.944	0.038	0.000	3.4	0.869	1	100%	79%
*hsa-mir-6888-5p*	0.628	0.097	0.197	3.6	0.438	0.819	50.7%	69.6%
*EDN1mRNA*	0.648	0.097	0.136	4.1	0.457	0.839	62.5%	64.5%
*EP300 mRNA*	0.707	0.089	0.037	2.3	0.533	0.832	81.3%	58.9%
*MAPK3 mRNA*	0.729	0.088	0.021	4.05	0.556	0.901	68.8%	58.9%
*TNF mRNA*	0.727	0.089	0.022	2.3	0.561	0.893	75.3%	58.9%

**Table 3 genes-12-01813-t003:** Multivariate regression analysis.

	B	S.E.	Sig.	Exp(B)	95% C.I. for EXP(B)	
					Lower	Upper
Age	0.007	0.029	0.803	1.007	0.952	1.066
*lncRNA RABGAP1L-DT-206*	−0.08	0.010	0.05	0.982	0.963	1.002
*has-miR-mir-6888-5p*	−0.021	0.015	0.174	0.979	0.951	1.009
*EDN1mRNA*	0.009	0.008	0.255	1.010	0.993	1.026
*EP300 mRNA*	−0.024	0.018	0.192	0.976	0.942	1.012
*MAPK3 mRNA*	−0.042	0.020	0.034	0.959	0.922	0.997
*TNF mRNA*	−0.045	0.020	0.025	0.956	0.918	0.994
ALT	−0.025	0.010	0.011	0.976	0.957	0.994
Constant	2.108	1.615	0.192	8.233		

## Data Availability

The data reported in this study are available on request from the corresponding authors.

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
