# Peer review of "Circulatory Endothelin 1-Regulating RNAs Panel: Promising Biomarkers for Non-Invasive NAFLD/NASH Diagnosis and Stratification: Clinical and Molecular Pilot Study"

_genes, 2021, doi:10.3390/genes12111813_

Round 1

Reviewer 1 Report

The authors analyzed and compared the expression of a serum RNAs panel in patients with fatty liver vs. healthy controls aiming to assess whether the STING signalling pathway had a predominant role in the former group. They found upregulation of the EDN1 regulating RNA expression in cases compared to healthy controls and concluded that such RNA panel could enable discriminate NAFLD cases from healthy controls and NAFLD from NASH cases. 

The topic is interesting and the experiments and sound. However, I have some concerns:

1- Cases and controls were identified by abdominal US and laboratory abnormalities. Although this is frequently used as a proxy in clinical routine to avoid liver biopsy in many patients, the definition of NAFLD still relies on at least 5% of hepatocytes with steatosis. Therefore, "NAFLD without steatosis" is conceptually incorrect. Moreover, US has a low negative predictive value for steatosis. 

2- How was NASH defined? I do not understand was the categorization done based on transient elastography.

3- My suggestions are as follows: first, repeat the analysis using just three groups: NAFLD/NASH without suspicion of advanced fibrosis (F3-F4, TE <8 kPa), NAFLD/NASH with suspicion of advanced fibrosis (TE =>kPA) and controls. Alternatively, if use NAFLD/NASH without advanced liver disease (TE<15kPa), NAFLD/NASH with cACLD (TE>15kPa), decompensated cirrhosis  (regardless of LS value in TE), and controls; secondly, define more accurately  how controls were selected (as mentioned, not having steatosis in the US is not equal of not having steatosis; what about other causes of liver disease such as virus, alcohol, etc...were they ruled out?); perhaps not labelling them as healthy controls but just controls...

Minor comments

4- English requires extensive copyediting

5- There are major formatting issues (e.g, references, spaces within paragraphs, low case using...)

6- The introduction can be shortened and ideas presented in a more orderly way, e.g., before mentioning the promising predictors of NAFLD or the role of RNAs explain what NAFLD is.

Author Response

Reviewer 1

Comments and Suggestions for Authors

The authors analyzed and compared the expression of a serum RNAs panel in patients with fatty liver vs. healthy controls aiming to assess whether the STING signalling pathway had a predominant role in the former group. They found upregulation of the EDN1 regulating RNA expression in cases compared to healthy controls and concluded that such RNA panel could enable discriminate NAFLD cases from healthy controls and NAFLD from NASH cases. 

The topic is interesting and the experiments and sound. However, I have some concerns:

  • Thanks very much for your valuable comments

  • Cases and controls were identified by abdominal US and laboratory abnormalities. Although this is frequently used as a proxy in clinical routine to avoid liver biopsy in many patients, the definition of NAFLD still relies on at least 5% of hepatocytes with steatosis. Therefore, "NAFLD without steatosis" is conceptually incorrect. Moreover, US has a low negative predictive value for steatosis. 
  • As suggested by the reviewer, NAFLD without steatosis has been deleted all though the MS according to your valuable comments and the groups have been rearranged as per your suggestions.

2- How was NASH defined? I do not understand was the categorization done based on transient elastography.

  • Transient elastography is an ultrasound-based study and also known as vibration-controlled transient elastography or Fibroscan which can measure controlled attenuation parameter (CAP). CAP which ranges from 100 to 400 decibels per meter (dB/m) can detect significant hepatic steatosis.
  • The optimal cut-off values of CAP for estimation of hepatic steatosis grades such as S1, S2, and S3 are ≥ 263dB/m, ≥ 281dB/m and ≥ 283dB/m respectively
  • CAP showed excellent diagnostic performance for differentiating presence and absence of hepatic steatosis by using a cutoff value of 241 dB/m in NAFLD but has limited value in evaluating grades of steatosis, especially in patients with high BMI (> 30 kg/m2)

  • Liu K, Wong VW, Lau K et al (2017) Prognostic value of controlled attenuation parameter by transient elastography. Am J Gastroenterol 112:1812–1823
  • Shin J, Kim MJ, Shin HJ et al (2019) Quick assessment with controlled attenuation parameter for hepatic steatosis in children based on MRI-PDFF as the gold standard. BMC Pediatr 19:112

3- My suggestions are as follows: first, repeat the analysis using just three groups:  NAFLD/NASH without suspicion of advanced fibrosis (F3-F4, TE <8 kPa), NAFLD/NASH with suspicion of advanced fibrosis (TE =>kPA) and controls.

Alternatively, if use NAFLD/NASH without advanced liver disease (TE<15kPa), NAFLD/NASH with cACLD (TE>15kPa), decompensated cirrhosis  (regardless of LS value in TE), and controls;

As suggested by the reviewer comments the statistical analysis were repeated in three groups

  1. NAFLD/NASH without suspicion of advanced fibrosis (F1-F2, TE <8 kPa),
  2. NAFLD/NASH with suspicion of advanced fibrosis (TE => 8kPA) and
  3.  

 Secondly, define more accurately  how controls were selected (as mentioned, not having steatosis in the US is not equal of not having steatosis; what about other causes of liver disease such as virus, alcohol, etc...were they ruled out?); perhaps not labeling them as healthy controls but just controls...

  • As suggested by the reviewer, controls selection criteria were clarified. Labeling of healthy control has been changed to controls
  • Furthermore, controls were age and sex matched to the study cases , negative viral markers and bilhariziasis , no alcoholic history intake, and normal liver function test, with confirmed normal imaging findings.

Table 1. Clinical and laboratory characteristics among the groups of the study.

variable

Group A NAFLD/NASH without suspicion of advanced fibrosis (F1-F2, TE <8 kPa), n=60

Group B NAFLD/NASH with  suspicion of advanced fibrosis (F3-F4, TE ≥8 kPa), n=40

          Group 3

Healthy control

n=100

P value

Sex

male

39(65%)

30(75%)

64(64%)

0.441

female

21(35%)

10(25%)

36(36%)

History of diabetes mellitus

0.00**

positive

49(81.7%)

34(85%)

42(42%)

negative

11(18.3%)

6(15%)

58(58%)

Body mass index (kg/m2) BMI

35.5±5.1

33.7±6.7

25.9 ±3.3

a0.00**

b 0.00**

c 0.134

Total cholesterol(mg/dL)

298.18±59.6

289.4±60.1

189 ±85.9

a0.00**

b 0.00**

c 0.545

LDLc(mg/dL)

209.9±49.5

199±60.4

136.18±66.3

a0.001**

b 0.00**

c 0.386

HDLc(mg/dL)

30.8.5±9.09

27.6±6.5

50.43±20.8

a0.00**

b 0.00**

c 0.328

Total triglycerides (mg/dL)

270.3±77.6

298.15±58.4

179.7±90.7

a0.00**

b 0.00**

c 0.106

albumin creatinie ratio

25.07±4.2

23.5±5.01

20.2±6.9

a0.00**

b 0.00**

c 0.207

AST(IU/L)

71.2±36.9

70.6±41.2

51±19.7

a0.00**

b 0.002**

c 0.993

ALT(IU/L)

46.3 ±25.2

59.7±44.8

34.3±16.4

a0.00**

b 0.007**

c 0.012*

Total bilirubin(mg/dL)

2.6 ±0.9

3 ±0.8

1.5 ±1.2

a0.00**

b 0.00**

c 0.057

Direct bilirubin (mg/dL)

1.5 ±0.66

1.7±0.69

0.88±0.39

a0.00**

b 0.00**

c 0.022*

Albumin(g/dL)

2.5 ±0.5

2.4 ±039

3.23±0.3

a0.00**

b 0.00**

c 0.065

Gamma glutammyl transferase (IU/L)

57.8±39.9

65.6±31.3

22.3±21.7

a0.00**

b 0.00**

c 0.243

alpha fetoprotein

180.5.9±439

359±433

18.0±31.27

a0.004**

b 0.00**

c 0.012*

Fasting blood glucose(mg/dL)

207.5±83.3

179.3 ±83.5

151.0 ±87

a0.000*

b 0.106

c 0.194

Glycated hemoglobin HbA1c (%)(%)

7.07±1.09

7.8±2.01

6.5±2.7

a0.27

b 0.008**

c 0.000*

HOMA IR

12.66±7.9

19.3±6.8

5.0±6.1

a0.000**

b 0.00**

c 0.000*

NAFLD Score

---

--

NAFLD Score < -1.455 = F0-F2

27 (45%)

0(0%)

NAFLD Score -1.455 – 0.675

33(65%)

2(5%)

NAFLD Score > 0.675 = F3-F4                

0(0%)

38(95%)

Fbrosis score

---

---

F0 to F1 Mild liver scaring

34(56.7%)

0(0%)

F2: Moderate liver scarring

26(46.3%)

0(0%)

F3: Severe liver scarring

0(0%)

29(72.5%)

F4: Advanced liver scarring (cirrhosis

0(0%)

11(27.5%)

steatosis grading

----

-------

S1 mild steatosis

15(25%)

0(0%)

4(10%)

36(90%)

0(0%)

S2  moderate stetosis

21(35%)

S3 severe steatosis

5(8.3%)

S4 non steatosis

19(31.7%)

One way Anova-test with post Hoc Turkey test was done to assess the differences among the study groups. Abbreviation: AST = aspartate transaminase, ALT = alanine transaminase, BMI = body mass index, FBS = fasting blood sugar, GGT= Gamma glutamyl transferase,, HDL-C = high density lipoprotein cholesterol, LDL-C = low density lipoprotein cholesterol,  TE=Transient elastography ,Kpa= kilopascal a control versus. Group A, b control versus. Group B , c Group A versus. Group B **p < 0.01 **p < 0.01; *p < 0.05.

Figure 2. Relative expression of circulatory RNAs panel among the study groups. (A) lncRNA RABGAP1L-DT-206, hsa-miR-6888-5p, (B) EP300 mRNA, EDN1 mRNA and (C) TNF mRNA and MAPK3 mRNA. & Statistically significant difference by post Hoc (Turkey) test. a control vs. Group A, b control vs. Group B, c Group A vs. Group B.

Table 2. Diagnostic Performance of the molecular parameters among the study groups.

Test Result Variable(s)

Area

Std. Error

Asymptotic Sig.

Asymptotic 95% Confidence Interval

cutoff

sensitivity

specificity

Lower Bound

Upper Bound

NAFLD/NASH vs. Control

lncRNA RABGAP1L-DT-206

.844

.031

.000

.82

.905

4.8

81%

83%

has-miR-mir-6888-5p

.916

.019

.000

.879

.953

1.97

91%

77%

EDN1mRNA

.797

.033

.000

.731

.862

1.85

87%

70%

EP300 mRNA

.839

.031

.000

.779

.900

2.15

83%

80%

MAPK3 mRNA

.871

.026

.000

.820

.921

2.65

88%

73%

TNF mRNA

.841

.031

.000

.781

.901

2.05

82%

81%

Combined RNAs

.888

.022

.000

.923

.845

3.25

91%

73%

AST

.653

.039

.000

57

.577

.729

55%

72%

ALT

.669

.039

.000

27

.593

.745

57%

73%

GGT

.806

.030

.000

39.5

.748

.864

66%

71%

Group A vs. Group B

lncRNA RABGAP1L-DT-206

.944

.038

.000

3.4

.869

1

100%

79%

hsa-mir-6888-5p

.628

.097

.197

3.6

.438

.819

50.7%

69.6%

EDN1mRNA

.648

.097

.136

4.1

.457

.839

62.5%

64.5%

EP300 mRNA

.707

.089

.037

2.3

.533

.832

81.3%

58.9%

MAPK3 mRNA

.729

.088

.021

4.05

.556

.901

68.8%

58.9%

TNF mRNA

.727

.089

.022

2.3

.561

.893

75.3%

58.9%

Figure 3. ROC curve analysis of (A) lncRNA RABGAP1L-DT-206, hsa-miR-6888-5p between NAFLD/NASH and controls, (B) EP300 mRNA, EDN1 mRNA between NAFLD/NASH and controls, (C) TNF mRNA, MAPK3 mRNA between NAFLD/NASH and controls (D) Combined RNAs panel between NAFLD/NASH and controls   (E) ALT, AST, GGT between NAFLD/NASH and controls, (F) lncRNA RABGAP1L-DT-206, hsa-miR-6888-5p, EP300 mRNA, EDN1 mRNA, TNF mRNA , and MAPK3 mRNA panel between Group A  and and group B.

Minor comments

4- English requires extensive copyediting

  • As suggested by the reviewer, English copyediting has been carried out

5- There are major formatting issues (e.g, references, spaces within paragraphs, low case using...)

  • As suggested by the reviewer, spaces, references, formatting issues have been corrected

6- The introduction can be shortened and ideas presented in a more orderly way, e.g., before mentioning the promising predictors of NAFLD or the role of RNAs explain what NAFLD is.

As suggested by the reviewer, the introduction has been shortened, rearranged as per your valuable comments. NAFLD introduction had been rearranged to precede NAFLD predictors.

Liver diseases cause two million deaths per year worldwide and thus they represent a universal health problem [[i]].Nonalcoholic fatty liver disease (NAFLD) is a progressive chronic liver disease characterized by excess fat accumulation in the liver. NAFLD can progress to nonalcoholic steatohepatitis (NASH) and, eventually, liver cirrhosis and hepatocellular carcinoma (HCC) worldwide [[ii]]. There are new promising predictors of NAFLD as combination of serum biomarkers that could help in early NASH diagnosis, but unfortunately with several well-known limitations. Although, magnetic resonance imaging-derived proton density fat fraction is considered the most accurate for fatty liver diagnosis. The main concern in clinical practice is early detection of NASH [[iii]].

NAFLD progression is attributed to many pathways e.g. oxidative stress, endoplasmic reticulum stress, and Toll like receptor -dependent release of cytokines [[iv]]. The liver acts as a primary immune cornerstone with various innate immune cells. Upon exposure to different stress signals, these innate immune cells become activated inducing the innate immune response and stimulating liver inflammation [[v]]. The cytosolic DNA induces the cyclic GMP-AMP synthase (cGAS)-stimulator of interferon genes (STING) pathway representing a critical signaling pathway of the innate immune system [[vi]]. Metabolic stress, such as a high-fat diet, obesity, and insulin resistance may stimulate cGAS and the STING-IRF3-mediated inflammation. Dysregulation of STING could inhibit free fatty acid induced inflammatory response, lipid accumulation, and hepatocellular damage [[vii]]. Lipotoxic stimulation affects downstream targets of cGAS-STING kinase, it induces the nuclear factor kappa beta ( NF-kB )signaling to produce proinflammatory cytokines that  activate macrophage to produce TGF-b1 and TNF-a which in turn stimulate hepatic stellate cells leading to  liver fibrosis in NASH [[viii]].

In NASH, chronic state of sterile inflammation is established due to the existingdamage associated molecular patterns (DAMPs) DAMPs; such as hepatocyte-mobility group-1 (HMGB1) and free fatty acids( FFAs) are endogenous molecules released from damaged cells that activate TLRs with subsequent inflammation, autophagy and apoptosis [[ix],[x]]. In cellular stress conditions, Hepatocyte mobility group (HMGB1) is moved from the nucleus into the cytoplasm, where it can affect intracellular processes such as autophagy. HMGB can act as pro-inflammatory mediators [[xi]]. Moreover, extracellular HMGB1 activates G protein coupled receptor (GPCR) and thus mediatingliver injury in NAFLD [[xii],[xiii]]. HMGB is linked to advanced glycation end products receptors (RAGE) that induces  inflammation in NAFLD via several GPCRs [[xiv]].

There is an interesting crosstalk between hepatocyte and liver macrophages. DAMP including GPCR bound toll-like receptor 4 TLR to activate nuclear factor (NF)-κB and TNFα secretion in Kupffer Cells(KC) [[xv]]. Recent study reported that the mitochondrial DNA acts as  stimulator of IFN genes (STING) in Kupffer Cells (KCs) to activate TNFα and IL-6 synthesis under the conditions of lipid overload [[xvi]].

The role of non-coding ncRNA in NAFLD progression has been discussed by numerous research groups e.g. miRNAs [[xvii],[xviii],[xix]] and lncRNAs [[xx],[xxi],[xxii]]. The integrated mRNA miRNAs lncRNA regulatory networks may provide new early diagnostic biomarkers and therapeutic strategies [[xxiii]].

 Based on this data, we constructed “mRNAs – miRNAs – lncRNAs” regulatory RNA network linked to hepatocyte –liver macrophage cross talk in NAFLD pathogenesis based on in biomarker filtration from public microarray databases. Then, we assessed NAFLD/NASH patients’ status versus control participants and measured the differential expression of the selected NAFLD specific RNA signature in sera samples.

  1. Xu Dongwei, Tian Yizhu, Xia Qiang, Ke Bibo. The cGAS-STING Pathway: Novel Perspectives in Liver Diseases. Frontiers in Immunology 2021,12.1569
  2. Bessone F, Razori MV, Roma MG. Molecular pathways of nonalcoholic fatty liver disease development and progression. Cell Mol Life Sci. 2019 Jan;76(1):99-128. doi: 10.1007/s00018-018-2947-0. Epub 2018 Oct 20. PMID: 30343320.
  3. Piazzolla VA, Mangia A. Noninvasive Diagnosis of NAFLD and NASH. Cells. 2020;9(4):1005. Published 2020 Apr 17. doi:10.3390/cells9041005
  4. [iv] Calzadilla Berlot L, Adams LA . The natural course of non-alcoholic fatty liver disease. Int J Mol Sci. (2016) 17:E774. https:// doi.org/10.3390/ijms17050774
  5. [v] Kazankov K, Jorgensen SMD, Thomsen KL, Moller HJ, Vilstrup H, George J,et al. The Role of Macrophages in Nonalcoholic Fatty Liver Disease and Nonalcoholic Steatohepatitis. Nat Rev Gastroenterol Hepatol (2019) 16:145–59. doi: 10.1038/s41575-018-0082-x
  6. [vi] Mao Y, Luo W, Zhang L, Wu W, Yuan L, Xu H, et al. Sting-Irf3 Triggers Endothelial Inflammation in Response to Free Fatty Acid-Induced Mitochondrial Damage in Diet-Induced Obesity. Arterioscler Thromb Vasc Biol (2017) 37:920–9. doi: 10.1161/ATVBAHA.117.309017
  7. [vii] Qiao JT, Cui C, Qing L, Wang LS, He TY, Yan F, et al. Activation of the STING-IRF3 Pathway Promotes Hepatocyte Inflammation, Apoptosis and Induces Metabolic Disorders in Nonalcoholic Fatty Liver Disease. Metabolism (2018) 81:13–24. doi: 10.1016/j.metabol.2017.09.010
  8. [viii] Luo X, Li H, Ma L, Zhou J, Guo X, Woo S-L, et al. Expression of STING is Increased in Liver Tissues From Patients With NAFLD and Promotes Macrophage-Mediated Hepatic Inflammation and Fibrosis in Mice.Gastroenterology (2018) 155:1971–84. doi: 10.1053/j.gastro.2018.09.010
  9. [ix] Ganz, M., Szabo, G. Immune and inflammatory pathways in NASH. Hepatol Int 7, 771–781 (2013). https://doi.org/10.1007/s12072-013-9468-6
  10. [x] Deng M, Scott MJ, Fan J, Billiar TR. Location is the key to function: HMGB1 in sepsis and trauma-induced inflammation. J Leukoc Biol. 2019;106:161–9.
  11. [xi] Li L, et al. Nuclear factor high-mobility group box1 mediating the activation of Toll-like receptor 4 signaling in hepatocytes in the early stage of nonalcoholic fatty liver disease in mice. Hepatology (Baltimore, MD). 2011;54:1620–30.
  12. [xii] Yuan, S., Liu, Z., Xu, Z. et al. High mobility group box 1 (HMGB1): a pivotal regulator of hematopoietic malignancies. J Hematol Oncol 13, 91 (2020). https://doi.org/10.1186/s13045-020-00920-3
  13. [xiii] Chen R, et al. Emerging role of high-mobility group box 1 (HMGB1) in liver diseases. Mol Med (Cambridge, Mass). 2013;19:357–66.
  14. [xiv] Chandrashekaran V, et al. HMGB1-RAGE pathway drives peroxynitrite signaling-induced IBD-like inflammation in murine nonalcoholic fatty liver disease. Redox Biol. 2017;13:8–19
  15. [xv] Garcia-Martinez I, Santoro N, Chen Y, Hoque R, Ouyang X, Caprio S, et al. Hepatocyte mitochondrial dNA drives nonalcoholic steatohepatitis by activation of tLR9. J Clin Invest. (2016) 126:859–64. doi: 10.1172/JCI83885
  16. [xvi] Yu Y, Liu Y, An W, Song J, Zhang Y, Zhao X. STING-mediated inflammation in kupffer cells contributes to progression of nonalcoholic steatohepatitis. J Clin Invest. (2019) 129:546–55. doi: 10.1172/JCI121842
  17. [xvii] Baffy, G. MicroRNAs in Nonalcoholic Fatty Liver Disease. J. Clin. Med. 2015, doi:10.3390/jcm4121953.
  18. [xviii] Gerhard, G.S.; DiStefano, J.K. Micro RNAs in the development of non-alcoholic fatty liver disease. World J. Hepatol. 2015.
  19. [xix] He, Z.; Hu, C.; Jia, W. miRNAs in non-alcoholic fatty liver disease. Front. Med. 2016.
  20. [xx] Chen, Y.; Huang, H.; Xu, C.; Yu, C.; Li, Y. Long non-coding RNA profiling in a non-alcoholic fatty liver disease rodent model: New insight into pathogenesis. Int. J. Mol. Sci. 2017, doi:10.3390/ijms18010021.
  21. [xxi] Grimaldi, B.; Bellet, M.M.; Katada, S.; Astarita, G.; Hirayama, J.; Amin, R.H.; Granneman, J.G.; Piomelli, D.; Leff, T.; Sassone-Corsi, P. PER2 controls lipid metabolism by direct regulation of PPARγ. Cell Metab. 2010, doi:10.1016/j.cmet.2010.10.005
  22. [xxii] Zhang, M.; Chi, X.; Qu, N.; Wang, C. Long noncoding RNA lncARSR promotes hepatic lipogenesis via Akt/SREBP-1c pathway and contributes to the pathogenesis of nonalcoholic steatohepatitis. Biochem. Biophys. Res. Commun. 2018, doi:10.1016/j.bbrc.2018.03.127.
  23. [xxiii] Wruck, W.; Graffmann, N.; Kawala, M.A.; Adjaye, J. Concise Review: Current Status and Future Directions on Research Related to Nonalcoholic Fatty Liver Disease. Stem Cells 2017, doi:10.1002/stem.2454.

Reviewer 2 Report

The authors in the work would want to develop novel ways to dragonize NAFLD/NASH vai non-invasive strategy. They have identified targets like EDN1/TNF/MAPK3/EP300/hsa-miR-6888-5p/lncRNA RABGAP1L-DT-206 and assessed their hypothesis in relation to STING pathway and NAFLD association. In addition, they have performed clinical studies on patients and corelated the marker levels and its NAFLD association. Please see the comments below.

  • The authors have done a neat work in their correlations. But still Tnfa being the major contributor for NAFLD/NASH progression is a known mechanism.
  • Some text editing needs to be done for proper formatting.
  • Figure format and representation is not consistent through the paper.

Minor revisions.

Author Response

Reviewer 2

The authors in the work would want to develop novel ways to dragonize NAFLD/NASH vai non-invasive strategy. They have identified targets like EDN1/TNF/MAPK3/EP300/hsa-miR-6888-5p/lncRNA RABGAP1L-DT-206 and assessed their hypothesis in relation to STING pathway and NAFLD association. In addition, they have performed clinical studies on patients and corelated the marker levels and its NAFLD association. Please see the comments below.

  • The authors have done a neat work in their correlations. But still Tnfa being the major contributor for NAFLD/NASH progression is a known mechanism.
  • Thanks very much for your valuable encouraging comments. In this study, we have tried to shed the light on molecular biology of the cGAS–STING pathway at genetic and epigenetic level that may enable to construct integrated panel connecting TNF with Endothelin 1- Regulating RNAs Panel to decrease false discovery rate associated with using TNF alone. Besides, this approach may help in the development of selective small-molecule inhibitors(siRNA,miRNA inhibitors) with the potential to target the cGAS–STING axis in NAFLD/NASH.
  • Some text editing needs to be done for proper formatting.
  • As suggested by the reviewer, English revision and text editing has been done all through MS.
  • Figure format and representation is not consistent through the paper.
  • As suggested by the reviewer, we have tried to make the figure format and representation consistent through the paper

Round 2

Reviewer 1 Report

The authors have satisfactorily addressed most of my remarks. However, I must insist that although CAP can indeed differentiate the severity of steatosis does not accurately predict the difference between having or not steatohepatitis. 

This manuscript is a resubmission of an earlier submission. The following is a list of the peer review reports and author responses from that submission.

Round 1

Reviewer 1 Report

The authors performed validation and validation experiments on public datasets for NAFLD/NASH diagnosis and stratification. I also tried to find the expression pattern of miRNA that can distinguish NAFLD/NASH well, and it is an interesting study, and an appropriate research method was presented.

Major issue:
1. It is necessary to show the overall expression pattern for the public dataset with a heatmap, etc.
2. The discussion is too long, as if reading a review paper. Interpretation of the results is not visible, so it must be rewritten in its entirety.

3. In the case of Figure 5, despite explaining the main point, it is not sincere. A figure appropriate to the quality of the "Genes" journal must be provided.

Minor issue:
1. The term "EDN1/TNF/MAPK3/EP300/hsa-miR-6888-5p/lncRNA RABGAP1L-DT-206 RNA panel" is too long. Appropriately abbreviated and presented in this paper.

2. The full name should come first, followed by the abbreviation. There are a lot of things that aren't. (Lines 68 and 72; HMGB1, lines 62 and 79; NF-kB, lines 80-81; KC, lines 60 and 68; FFA, lines 56-57 and 81; STING, etc)

3. Clearly indicate the p-value from p=0.00 to p<0.001.

4. Significant figures should be unified. (e.g. 5±6.1 > 5.0±6.1)

5. Line 108: TNF, MAPK3, EP300, and EDN1 mRNA

6. Visualization through the boxplot in Figure 1 is an appropriate strategy. But is it need labeling for outliers? Delete the label except for the top three.

7. The ROC curves and sensitivity and specificity data in Table 2 and Figure 2A-E are reasonable. However, the background grid in figure 2 is unnecessary.

8. The correlation analysis in Figure 4 is reasonable. However, a regression line is required in the figure, it is desirable that the correlation coefficient be provided in the figure, and the background grid is unnecessary.

9. Provide the NCBI GEO accession number in the method.

10. In the case of Figure 5, despite explaining the main point, it is not sincere. A figure appropriate to the quality of the "Genes" journal must be provided.

11. The supplementary figure is not organized, it is just a simple list of the contents searched by the NCBI. It should be presented in an appropriate and organized form.

Author Response

Reviewer 1

Comments and Suggestions for Authors

The authors performed validation and validation experiments on public datasets for NAFLD/NASH diagnosis and stratification. I also tried to find the expression pattern of miRNA that can distinguish NAFLD/NASH well, and it is an interesting study, and an appropriate research method was presented.

Major issue:
1. It is necessary to show the overall expression pattern for the public dataset with a heatmap, etc.

  • As suggested by the reviewer, more details about GEO dataset Dataset: GSE89632 with analysis of DEG by repository GEO2R and Gene ontology (GO) enrichment and pathway analyses of the retrieved 9969 DEGs were performed using Enrichr(supplementary table 1,2,3, figure 1)
  • As suggested by the reviewer, the overall expression pattern for the public dataset has presented with a heatmap

Figure 1A: Heat map of differentially expressed genes in GSE89632. Figure1B: Top 10 items of Gene Ontology (Biological processes) for the retrieved DEGs according to p value obtained from Enrichr. Figure 1C: Top 10 items of KEGG pathways for the retrieved DEGs according to adjust p value obtained from Enrichr.

  1. The discussion is too long, as if reading a review paper. Interpretation of the results is not visible, so it must be rewritten in its entirety.
  • As suggested by the reviewer, The discussion has been summarized and more interpretation of the results has been added
  1. In the case of Figure 5, despite explaining the main point, it is not sincere. A figure appropriate to the quality of the "Genes" journal must be provided.

As suggested by the reviewer, Figure 5 has been modified

Figure 6 summary and schematic presentation of the study findings.

Minor issue:
1. The term "EDN1/TNF/MAPK3/EP300/hsa-miR-6888-5p/lncRNA RABGAP1L-DT-206 RNA panel" is too long. Appropriately abbreviated and presented in this paper.

As suggested by the reviewer, EDN1/TNF/MAPK3/EP300/hsa-miR-6888-5p/lncRNA RABGAP1L-DT-206 RNA panel has been abbreviated as EDN1-regulating RNAs Panel

  1. The full name should come first, followed by the abbreviation. There are a lot of things that aren't. (Lines 68 and 72; HMGB1, lines 62 and 79; NF-kB, lines 80-81; KC, lines 60 and 68; FFA, lines 56-57 and 81; STING, etc)

As suggested by the reviewer, the full names had been added at first, followed by the abbreviation in the required lines

  1. Clearly indicate the p-value from p=0.00 to p<0.001.

As suggested by the reviewer, the p-value has been clearly indicated in the methodology from p=0.00 to p<0.001

  1. Significant figures should be unified. (e.g. 5±6.1 > 5.0±6.1)

As suggested by the reviewer, 5±6.1 has been unified

  1. Line 108: TNF, MAPK3, EP300, and EDN1 mRNA

As suggested by the reviewer, mRNAs have been rearranged as follows; EDN1 mRNA, EP300, MAPK3, and TNF.

  1. Visualization through the boxplot in Figure 1 is an appropriate strategy. But is it need labeling for outliers? Delete the label except for the top three.

As suggested by the reviewer, Figure 1 has been modified

  1. The ROC curves and sensitivity and specificity data in Table 2 and Figure 2A-E are reasonable. However, the background grid in figure 2 is unnecessary.

As suggested by the reviewer, Figure 2 has been modified

  1. The correlation analysis in Figure 4 is reasonable. However, a regression line is required in the figure, it is desirable that the correlation coefficient be provided in the figure, and the background grid is unnecessary.

As suggested by the reviewer, Figure 4 has been modified

  1. Provide the NCBI GEO accession number in the method.

As suggested by the reviewer, NCBI GEO accession number has been added, Dataset: GSE89632

  1. In the case of Figure 5, despite explaining the main point, it is not sincere. A figure appropriate to the quality of the "Genes" journal must be provided.

As suggested by the reviewer, Figure 5 has been modified

  1. The supplementary figure is not organized, it is just a simple list of the contents searched by the NCBI. It should be presented in an appropriate and organized form.

As suggested by the reviewer, The supplementary materials have been modified as per your valuable comments to address the detailed gene expression of Dataset: GSE89632 including more tables about DEG  and GO. The supplementary in the submitted version 1 has been incorporated as verification step following GEO data set analysis

Reviewer 2 Report

In this manuscript, Reda Albadawy and colleagues selected four mRNAs (EDN1, TNF, MAPK3, EP300), one microRNA (miR-6888-5p) and one long ncRNA (RABGAP1L-DT-206) and measured their circulating levels in the serum of NAFLD and NASH patients, and healthy controls. Although the data reported are novel, this work needs to be improved in many parts, especially the introduction and presentation of results. The figures are not appreciable (legends and * too small, graphs titles unclear) and the legends are incomplete. The discussion can be enriched with comments on the data obtained.

Here, some major comments:

- it is not clear why the authors selected EDN1, TNF, MAPK3, and EP300 as genes implicated in NASH. In my opinion, the research design of the selection of these molecules must be elucidated. In this regard, the authors should include all the details of data acquisition from GEO as well as the meta-analysis of these data. There are several studies in the literature based on public data, I strongly recommend consulting them.

- Figures S3 and S4 are necessary to verify the bioinformatic analysis that has been made. In addition, there are no published data on miR-6888-5p. In my opinion, the Gene Ontology (GO) analysis (i.e. using DAVID) of the predictive miRNA targets can be very informative on the biological role of this miR.

- microRNAs mainly regulate gene expression by inducing the silencing of mRNA targets. Thus, it is expected that miR-6888-5p negatively regulates EDN1, TNF, MAPK3, and EP300. However, the data obtained showed the up-regulation of these mRNAs as well as miR-6888 in the serum of NAFLD patients. The authors should comment on this point and consider performing the analysis of public data obtained in liver tissues.

- I suggest changing the formula “… RNA panel” also in the title. The potential diagnostic value of a “RNAs panel” could be evaluated by a ROC curve analysis with the combinations of RNA molecules (for example, gene coding transcripts and non-coding RNAs).

Author Response

Reviewer 2

In this manuscript, Reda Albadawy and colleagues selected four mRNAs (EDN1, TNF, MAPK3, EP300), one microRNA (miR-6888-5p) and one long ncRNA (RABGAP1L-DT-206) and measured their circulating levels in the serum of NAFLD and NASH patients, and healthy controls. Although the data reported are novel, this work needs to be improved in many parts, especially the introduction and presentation of results. The figures are not appreciable (legends and * too small, graphs titles unclear) and the legends are incomplete. The discussion can be enriched with comments on the data obtained.

As suggested by the reviewer, The introduction, method , results presentation and discussion has been modified. Moreovere, we have tried to modify and improve the figure.

Here, some major comments:

- it is not clear why the authors selected EDN1, TNF, MAPK3, and EP300 as genes implicated in NASH. In my opinion, the research design of the selection of these molecules must be elucidated. In this regard, the authors should include all the details of data acquisition from GEO as well as the meta-analysis of these data. There are several studies in the literature based on public data, I strongly recommend consulting them.

  • As suggested by the reviewer, The rationale for selection of EDN1, TNF, MAPK3, and EP300 from GEO dataset has been clearly indicated in the methodology and result section. In addition, the details of GEO Dataset: GSE89632 and metanalysis of theses data has been added (supplementary table 1, 2 and figure 1).
  • Results
  • Retrieval of differentially expressed mRNAs(DEG) from GEO data set :

By normalization and analysis of the microarray dataset, a number of DEGs were identified in GSE33814 (Figure 1 , Supplementary Table S1). The GSE33814 dataset contained 9969 DEGs were identified based on the appropriate cut-off . We used the Enrichr database for functional enrichment analysis DEGs between NASH, steatosis and normal groups(Figure 1 , Supplementary Table 2). Then key genes EDN1, EP300, MAPK3, and TNF were selected for the targeted network and validated by other GEO datasets and  other  public databases to be related to STING signaling , cytokine response and NAFLD/NASH pathogenesis (Supplementary Figure S1, S2). These selected genes were imported into string database for PPI network construction (Figure S2 ). Additionally, We used DAVID Functional enrichment tool(https://david.ncifcrf.gov/tools.jsp, accessed on 15-10-2021)  that revealed that validated  biological function of EP300 and MAPK3 in cytokine response and the molecular function of the 4 selected genes in regulation of RNA transcription and MAP kinase signaling (Figure S.3)  Then, the targeted miRNA were selected from miRWalk 3.0, namely; has-miR-6888-5p, could interact with 4 differentially expressed mRNAs identified above (Supplementary Figure S3). lately,  we used mirwalk2 to predict the interaction between lncRNAs and miRNAs RABGAP1L-DT-206, was screened and interacting with the retrieved miRNA (Supplementary Figure S4).Finally, (EDN1/TNF/MAPK3/EP300/hsa-miR-6888-5p/lncRNA RABGAP1L-DT-206 RNAs panel was constructed.  

  • Methodology
  • 1. Biomarker Filtration of mRNA-miRNA-lncRNA Panel From Public Microarray Database
  • The candidate genes of the present study were acquired from the GEO database (ncbi.nlm.nih.gov/geo/, accessed on 15-10-2021) [[i]]. The search was restricted to homo sapiens and the experimental articles which contained whole-gene expression data that differentiate between the NASH and normal control groups were included. As a result, The GSE89632 dataset was obtained [[ii]]. Detailed parameters of the dataset are presented in Supplementary Table I. The GSE89632 dataset represents a cross-sectional study that used hepatic gene expression on  Illumina Microarray  and compared 20 patients with simple steatosis, 19 with nonalcoholic steatohepatitis (NASH), and 24 healthy controls (HC).   Subsequently, microarray data from the GSE89632 was submitted to the online database repository GEO2R (https://www.ncbi.nlm.nih.gov/geo/geo2r/, accessed on 15-10-2021) to identify differentially expressed genes (DEGs) among the groups (Supplementary Table 1). A p-value of <0.05 was considered to indicate a statistically significant difference. Finally, Gene ontology (GO) enrichment and pathway analyses of the retrieved 9969 DEGs were performed using Enrichr (http://amp.pharm.mssm.edu/Enrichr, accessed on 15-10-2021) [[iii]]. The result was summarized in (Supplementary Table 2.) and figure 1.
  • Afterwards , The integrated RNA panel was filtered and verified in three steps from other GEO  datasets and other microarray databases:
  • (i) Endothelin 1 (EDN1), E1A Binding Protein P300 (EP300) , Mitogen-Activated Protein Kinase 3(MAPK3) and Tumor Necrosis factor alpha (TNFα) were verified based upon their correlation to STING related cytokine response and strong implication in NASH pathogenesis. The chosen messenger RNAs were also verified for their gene ontology and expression by using several public microarray databases; QuickGO (https://www.ebi.ac.uk/QuickGO/), and National Center of Biotechnology Information Gene (https://www.ncbi.nlm.nih.gov/gene ) (S1. Fig) and by literature reviews [16–22] to be related to cytokine and Cytosolic DNA-sensing pathway STING signaling pathway by KEGG (https://www.genome.jp/kegg/) (S2. Fig). The four chosen genes were uploaded into the Search Tool for the Retrieval of Interacting Genes (STRING; version 11.0; http://stringdb.org) database to assess protein-protein cross talk (Fig. S2) and DAVID functional enrichment tool to highlight their gene ontology in NAFLD/NASH  progression (Fig S2).
  • (ii) We used miRWalk 3.0 (http://mirwalk.umm.uni-heidelberg.de/) to select miRNA which interact with the four selected mRNAs. It revealed that miR-6888-5p could target the selected mRNAs (Fig. S3). Also, miRPath database version 2 (https://mpd.bioinf.uni-sb.de/mirna.html?mirna=hsa-miR-6888-5p&organism=hsa) was used to carry out pathway enrichment analysis of miR-6888-5p that was linked to regulation of gene expression, RNA polymerase and cell morphogenesis (S3. Fig).
  • (iii) We used miRWalk 2.0; miRNA:ncRNA target tool (http://zmf.umm.uni-heidelberg.de/apps/zmf/mirwalk2/mir-mir-self.html) to predict the interaction between miRNA and lncRNA. RABGAP1L-DT 206(ENSG00000227373, ENST00000454467.1) was identified to be interacting with the chosen miR-6888-5p and that was validated through Clustal Omega tool of The European Bioinformatics Institute (EMBL-EBI) (https://www.ebi.ac.uk/Tools/msa/clustalo/) (S4. Fig).
  • All in all, (EDN1, EP300, MAPK3 & TNFα) - (miR-6888-5p) - (RABGAP1L-DT-206) RNA panel was constructed.

- Figures S3 and S4 are necessary to verify the bioinformatic analysis that has been made. In addition, there are no published data on miR-6888-5p. In my opinion, the Gene Ontology (GO) analysis (i.e. using DAVID) of the predictive miRNA targets can be very informative on the biological role of this miR.

  • As suggested by the reviewer, Figures S3 and S4 has been added
  • S3 Figure: Validation of the interaction between the selected mRNAs and the retrieved has-miR-6888-5p from mirWalk database and Gene card database

  • S4 Figure: Validation of the interaction between the retrieved hsa-miR-6888-5p and lncRNA RABGAP1L-DT-206
  • As suggested by the reviewer, the Gene Ontology (GO) analysis (i.e. using DAVID) has been added according to your valuable comments in the method, results and supplementary figures

  •  

- microRNAs mainly regulate gene expression by inducing the silencing of mRNA targets. Thus, it is expected that miR-6888-5p negatively regulates EDN1, TNF, MAPK3, and EP300. However, the data obtained showed the up-regulation of these mRNAs as well as miR-6888 in the serum of NAFLD patients. The authors should comment on this point and consider performing the analysis of public data obtained in liver tissues.

  • As suggested by the reviewer, the role of miRNA in the activation of target genes has been addressed in the discussion.
  • In the current study, we have assessed the expression hsa-miR-6888-5p as a retrieved epigenetic activator of the EDN1/TNF/MAPK3/EP300/panel in agreement with the recent evidence that miRNAs could interact with the promoter and enhance gene expression through man miRNA-induced RNA activation [[iv], [v]].

- I suggest changing the formula “… RNA panel” also in the title. The potential diagnostic value of a “RNAs panel” could be evaluated by a ROC curve analysis with the combinations of RNA molecules (for example, gene coding transcripts and non-coding RNAs).

As suggested by the reviewer,RNA panel  has been changed  to RNAs panel. Also ROCcurve for combined RNAs has been added to Figure 3 and table 2

ROC Combined RNAs panel between NAFLD and healthy controls  

ROC curve between NASH and NAFLD without steatosis

between NASH and NAFLD with simple steatosis

Table 2.Dianostic Performance of the molecular parameters among the study groups.

Test Result Variable(s)

Area

Std. Error

Asymptotic Sig.

Asymptotic 95% Confidence Interval

cutoff

sensitivity

specificit

Lower Bound

Upper Bound

NAFLD vs. Healthy control

lncRNA RABGAP1L-DT-206

.844

.031

.000

.783

.905

4.8

81%

83%

has-miR-mir-6888-5p

.916

.019

.000

.879

.953

1.97

91%

77%

EDN1mRNA

.797

.033

.000

.731

.862

1.85

87%

70%

EP300 mRNA

.839

.031

.000

.779

.900

2.15

83%

80%

MAPK3 mRNA

.871

.026

.000

.820

.921

2.65

88%

73%

TNF mRNA

.841

.031

.000

.781

.901

2.05

82%

81%

Combined RNAs

.888

.022

.000

.923

.845

3.25

91%

73%

AST

.653

.039

.000

57

.577

.729

55%

72%

ALT

.669

.039

.000

27

.593

.745

57%

73%

GGT

.806

.030

.000

39.5

.748

.864

66%

71%

NASH vs. NAFLD

lncRNA RABGAP1L-DT-206

.862

.042

.000

20.5

.779

.944

87.7%

81.8%

hsa-mir-6888-5p

.877

.043

.000

12.8

.793

.961

90%

80%

EDN1mRNA

.818

.049

.000

7.2

.723

.913

87.7%

77.4%

EP300 mRNA

.750

.052

.000

9.5

.649

.852

78.5%

66.4%

MAPK3 mRNA

.811

.050

.000

9.1

.713

.910

86%

83%

TNF mRNA

.815

.046

.000

10

.725

.905

80%

81%

combined RNAs

.864

.044

.000

  .777

.950

20.5

87.7

81.9

NASH vs. simple steatosis

lncRNA RABGAP1L-DT-206

.862

.042

.000

20.5

.779

.944

87.7%

81.8%

hsa-mir-6888-5p

.877

.043

.000

12.8

.793

.961

90%

80%

EDN1mRNA

.818

.049

.000

7.2

.723

.913

87.7%

77.4%

EP300 mRNA

.750

.052

.000

9.5

.649

.852

78.5%

66.4%

MAPK3 mRNA

.811

.050

.000

9.1

.713

.910

86%

83%

TNF mRNA

.815

.046

.000

10

.725

.905

80%

81%

combined RNAs

.756

.081

.002

.596

.915

30.05

78.5%

69%

[i] R. Edgar, M. Domrachev, and A. E. Lash, “Gene Expression Omnibus: NCBI gene expression and hybridization array data repository,” Nucleic Acids Res., 2002, doi: 10.1093/nar/30.1.207

[ii] B. M. Arendt et al., “Altered hepatic gene expression in nonalcoholic fatty liver disease is associated with lower hepatic n-3 and n-6 polyunsaturated fatty acids,” Hepatology, 2015, doi: 10.1002/hep.27695.

[iii] M. V. Kuleshov et al., “Enrichr: a comprehensive gene set enrichment analysis web server 2016 update,” Nucleic Acids Res., 2016, doi: 10.1093/nar/gkw377.

[iv] Vaschetto LM. miRNA activation is an endogenous gene expression pathway. RNA Biol. 2018;15(6):826-828. doi: 10.1080/15476286.2018.1451722. Epub 2018 Apr 3. PMID: 29537927; PMCID: PMC6152443.

[v] Ramchandran R, Chaluvally-Raghavan P. miRNA-Mediated RNA Activation in Mammalian Cells. Adv Exp Med Biol. 2017;983:81-89. doi: 10.1007/978-981-10-4310-9_6. PMID: 28639193.

Round 2

Reviewer 1 Report

Dear authors,
Thank you for editing the manuscript.
Some parts were correlated to the pointed out.
However, some parts were not corrected.
(Discussion section, figure, etc)
Significant figures should be unified. (e.g. 5±6.1 > 5.0±6.1)
In the boxplot, delete the label except for the top three.
It presents a good topic and is a good thesis to study the characteristics of NAFLD/NASH.

Reviewer 2 Report

As I already suggest in Report 1, the paper deals with a very interesting topic into account the importance to identify circulating biomarkers for NAFLD/NASH diagnosis. In the revised version of the manuscript, the authors reported the heat map of differentially expressed genes in a specific dataset from GEO (GSE89632) and analyzed the diagnostic potential value of all selected transcripts by ROC curve. However, authors must provide compelling evidence and rationale behind measuring the selected transcripts. To this aim:

- a scatter plot should be used to illustrate the levels of the selected mRNAs in NASH and normal tissues for each GSE data set (since the authors reported that these genes were validated by other GEO datasets – line 158);

- functional annotation analysis by DAVID or EnrichR should support the selection of EDN1, EP300, MAPK3, and TNF as NAFLD/NASH-related genes. If so, this point is not evidenced in the results section and a complete table reporting the gene annotation results could help to detect all the interesting GO terms associated with liver inflammation;

- the authors selected miR-6888-5p as a regulator of EDN1, EP300, MAPK3, and TNF and they evaluated its circulating levels. However, the bioinformatics analysis to predict the interaction miRNA::mRNA was conducted on a different miRNA (miR-6886-5p, Fig.S3). This is a fundamental step for the construction of a miRNA-mRNA signature. In addition, I suggest using more databases (e.g. targetscan, miRdb…)

Finally, the text is often incongruent with many oversights and the editing of English is required. The figures are still not appropriate:

- Fig.2 and Fig.4: the histograms are not clear, the symbols of significance are not reported. EDN1 mRNA levels are reported in Fig.2A and Fig.2B, but there aren't the levels of lncRNA. I recommend the scatter plot for this type of representation.

- The supplementary figures are merely screenshots of databases pages.